# Effect of Heavy Ion ^12^C^6+^ Radiation on Lipid Constitution in the Rat Brain

**DOI:** 10.3390/molecules25163762

**Published:** 2020-08-18

**Authors:** Bo Li, Chu Han, Yuanyuan Liu, Nafissa Ismail, Kevin Smith, Peng Zhang, Zixuan Chen, Rongji Dai, Yulin Deng

**Affiliations:** 1Beijing Key Laboratory for Separation and Analysis in Biomedicine and Pharmaceuticals, School of Life Science, Beijing Institute of Technology, Beijing 100081, China; zp989dream@163.com (P.Z.); zx-chen@bit.edu.cn (Z.C.); 2Advanced Research Institute of Multidisciplinary Science, Beijing Institute of Technology, Beijing 100081, China; 3120181225@bit.edu.cn (C.H.); 3120181246@bit.edu.cn (Y.L.); 3School of Chemistry and Chemical Engineering, Beijing Institute of Technology, Beijing 100081, China; 4Neuroimmunology, Stress and Endocrinology (NISE) Lab, School of Psychology, Faculty of Social Science, University of Ottawa, Ottawa, ON K1N 6N5, Canada; Nafissa.Ismail@uottawa.ca (N.I.); ksmit006@uottawa.ca (K.S.); 5Brain and Mind Research Institute, University of Ottawa, Ottawa, ON K1N 6N5, Canada

**Keywords:** heavy ion radiation, lipidomics, brain damage, UPLC–MS, endoplasmic reticulum stress

## Abstract

Heavy ions refer to charged particles with a mass greater than four (i.e., alpha particles). The heavy ion irradiation used in radiotherapy or that astronauts suffer in space flight missions induces toxicity in normal tissue and leads to short-term and long-term damage in both the structure and function of the brain. However, the underlying molecular alterations caused by heavy ion radiation have yet to be completely elucidated. Herein, untargeted and targeted lipidomic profiling of the whole brain tissue and blood plasma 7 days after the administration of the 15 Gy (260 MeV, low linear energy (LET) = 13.9 KeV/μm) plateau irradiation of disposable ^12^C^6+^ heavy ions on the whole heads of rats was explored to study the lipid damage induced by heavy ion radiation in the rat brain using ultra performance liquid chromatography-mass spectrometry (UPLC–MS) technology. Combined with multivariate variables and univariate data analysis methods, our results indicated that an orthogonal partial least squares discriminant analysis (OPLS–DA) could clearly distinguish lipid metabolites between the irradiated and control groups. Through the combination of variable weight value (VIP), variation multiple (FC), and differential (p) analyses, the significant differential lipids diacylglycerols (DAGs) were screened out. Further quantitative targeted lipidomic analyses of these DAGs in the rat brain tissue and plasma supported the notion that DAG 47:1 could be used as a potential biomarker to study brain injury induced by heavy ion irradiation.

## 1. Introduction

In biomedical research, heavy ion beam currents are used to investigate the biological effects of ground-based simulated cosmic rays. They are also commonly used in radiotherapy to treat cancer patients [1]. Unlike X-ray and gamma radiation, heavy ions have a reverse depth dose distribution and a high relative biological effect (RBE), making them ideal tools for the treatment of head, neck, and central nervous system tumors [2]. Furthermore, cranial radiotherapy remains the mainstay of palliative treatment in patients with multiple symptomatic brain metastases [3]. Radiotherapy not only targets tumors, but also affects the healthy brain tissue surrounding the tumor. However, research on the effect of radiotherapy on healthy brain tissue is limited. Studies have shown that heavy ion radiation causes structural and functional damage in the central nervous system (CNS), regardless of low or high low linear energy (LET) irradiation [4,5]. Clinically, CNS damage caused by whole brain irradiation is divided into acute (≤7d) and early (up to months) and late-delayed stages (several months to years) [6]. Brain histopathology at the late-delayed stage following medium or high irradiation with heavy ions showed irreversible cerebral spinal cord disorders, including vascular injury and white matter necrosis [7,8]. The behavioral consequences of spinal cord disorders include cognitive dysfunction and abnormal walking patterns [7,9]. In contrast, the pathological and behavioral consequences of the acute stage following irradiation are more subtle [7]. Non-invasive, early-onset biomarkers have been developed in response to heavy ion radiation damage-mediated cognitive decline.

Heavy ion irradiation damages DNA, proteins, lipids, and the cell structure and function, leading to the production of free radicals. Following ionizing radiation, lipid-derived free radicals (R·) generate lipid peroxide radicals (ROO·) in the presence of oxygen, causing damage to lipid-rich cell membrane structures Harmful substances enter the cells to attack DNA and proteins, resulting in unwanted gene translocation, chromosomal aberrations, the inactivation and degradation of proteins, and enzymatic activity [10,11]. The lipids that make up the cell membrane affect the shape, curvature, and fluidity of the cell and regulate exocytosis and endocytosis. Lipids also regulate the enzymatic activity and protein trafficking at the synaptic site of neurons by interacting with specific proteins and signaling complexes [12]. In addition, lipids also regulate the release of neurotransmitters and the propagation of signal transduction [13].

Lipid metabolism is of particular interest, due to its high concentration in the CNS. The importance of lipids in cell signaling and tissue physiology has been demonstrated in many CNS disorders and injuries that involve deregulated metabolism [14]. In the human, chimpanzee, rhesus monkey, and mouse, the concentration of lipids in the brain has evolved faster than in non-neural tissues [15]. Exceptional accelerated evolution in the prefrontal cortex revealed that lipids play an important role in cognitive function [15]. Furthermore, cholesterol synthesis or lipoprotein transport impairment reduce synaptic plasticity, neuronal survival and performance, and overall cognitive function [16].

Focusing on the comprehensive identification and quantification of lipids, lipidomics delivers biological information through dynamic changes in lipids [17,18]. In recent years, lipidomics have also been used widely in the diagnosis and treatment of neurodegenerative diseases and strokes [19,20]. For example, lipidomics are used to carry out the detailed and systematic characterization of components of brain phospholipids in Parkinson’s disease (PD) mouse models, in which brain phospholipids’ distribution was age-dependent [21]. Lipid changes lead to alterations in post-translational modifications of PD-related proteins, including α-Syn [21]. Lipidomic analysis identified 10 biomarkers, including 8 phosphatidylcholines, from peripheral blood plasma to predict the phenotypic transformation from normal to amnestic mild cognitive impairment [22]. Additionally, the onset of Alzheimer’s disease was predicted within 2–3 years with a >90% accuracy [22]. This study also reveals that lipid damage studies are useful for detecting behavioral changes.

We previously constructed the rat model of brain-localized 15 Gy carbon ion radiation (plateau region) and found that rats can survive for more than three months after irradiation [23]. We found that brain irradiation severely affects the peripheral immune system, even long after irradiation. Therefore, in order to study the effect of heavy ion radiation on lipid composition, as well as to provide insight into the mechanism of brain damage, a thorough lipidomic study of the whole rat brain 7 days (acute phase) after it was exposed to 15 Gy of a ^12^C^6+^ heavy ion beam (260 MeV, LET = 13.9 KeV/μm) was carried out using a UPLC–MS-based lipidomic method. The differential lipids were screened from untargeted lipidomics, and additional quantitative analyses were used to confirm the potential lipid biomarkers in the damaged brain tissue and blood plasma. Potential lipid targets DAG 47:1 were found. The mechanism of damage on lipids was hypothesized and discussed through discovering and comparing differential lipids. To the best of our knowledge, this is the first lipidomic study of brain damage induced by heave heavy ion irradiation on a molecular level. Our results indicate that lipidomics is a valuable approach to identify potential detrimental pathways in the early stage post-irradiation. Significant differential lipids can be potentially used as biomarkers to measure brain injuries induced by heavy ion irradiation.

## 2. Results

### 2.1. The Effect of Heavy Ion Irradiation on the Brain Weight of Rats

To examine the effect of heavy ion irradiation on brain damage, 15 Gy (260 MeV, LET = 13.9 KeV/μm) of ^12^C^6+^ irradiation at a rate of 0.5 Gy/min was applied to whole heads of Wistar wild-type rats. The plateau region of heavy ion beam and X-rays belongs to low linear energy (LET) radiation [24]. The radiation biological effect produced by the plateau region of the heavy ion beam is much lower than the radiation biological effect produced in or near the Bragg peak area [2]. On the 7th day post-irradiation, the rats were sacrificed and the whole brains were taken out for further lipidomic analysis. Three samples from each group were randomly chosen and the weight of the whole brain was measured. As shown in Table 1, the average weight of the brains in the irradiated rats was 1.529 g per rat, which was 11.8% less compared to the brain weights of the control group, which were 1.733 g per rat on average. On the 7th day after irradiation, the average body weight of the rats in the irradiation group was about the same as their weight before irradiation; however, the average body weight of the rats in the control group increased about 14% (Appendix A).

### 2.2. Pathological Analysis of the Rat Brain Tissue after Heavy Ion Irradiation

Radiation-induced cognitive dysfunction frequently occurs in the hippocampus and prefrontal cortex [25,26]. In order to determine the effects of ^12^C^6+^ irradiation on the rat brain, we randomly selected a rat’s left hemisphere from each group to make paraffin section and H&E staining analyses (see Appendix A). Compared with the control group, the neuron nuclei in the prefrontal cortex of the irradiated rats showed vacuolation (black arrow in Appendix A), but no inflammatory infiltration could be observed. The results of the pathological analysis of the hippocampus, another sensitive area of brain radiation damage, showed that the neurons were abundant and closely arranged, with a normal neuronal morphology and obvious nucleolus in both the control and irradiated group. The above results indicated that the acute period after 15 Gy ^12^C^6+^ irradiation did cause morphology changes in cells, although this was not obvious in the rat brain after 7 days.

### 2.3. The Effect of Heavy Ion Irradiation on the Lipid Composition in the Rat Brain

Lipid composition and metabolism is of particular interest in studying the pathology of the brain. We studied the lipid composition to explore the mechanism of brain damage induced by heavy ion irradiation. Brain tissue samples were collected on the 7th day post-irradiation. We used comparative lipidomics based on UPLC–MS technology to analyze the central lipid composition to capture early changes in the lipidomic profiles. The lipid mixtures were separated by ultra-performance liquid chromatography. Different types of lipid molecules were identified by mass spectrometry in both the positive and negative ion modes, respectively. We observed differences in the total ion chromatogram (TIC) peaks, suggesting that exposure to heavy ion irradiation significantly altered the central lipidomic profiles.

To identify lipidomic differences in brain tissue between the irradiated and control group, the unsupervised principal component analysis (PCA) and supervised orthogonal partial least squares discrimination analysis (OPLS–DA) were used to examine the distribution of lipids in the two groups. To investigate the global metabolic variations, the observations were analyzed using PCA (Figure 1A). It could be seen from the PCA score chart that the control group and the radiation group partially overlapped. The separation of samples using principal component analysis was not satisfactory. Therefore, we chose to proceed using OPLS–DA to measure the group separation. In addition to explaining the overall differences between classes, OPLS–DA is an extended form of partial least squares-discriminant analysis (PLS–DA) that separates predictive and non-predictive variation. The score plots from the OPLS–DA model displayed a clear differentiation between two groups, as shown in Figure 1B. The quality and effectiveness of the constructed model were evaluated using the goodness of fit (R^2^) and the predictability (Q^2^). In order to prevent false positives, the model was tested with 100 response ranking tests. The result obtained with R^2^ was 0.981, and the Q^2^ was 43.7%.

Molecules with an FC > 1.5, VIP > 1, and *p* < 0.05 were considered as statistically significant changes in lipids induced by heavy ion radiation exposure. The processing of LC–MS data resulted in a total of 3031 detected lipids. Among them, 2152 and 879 lipids were detected in the positive and negative ion modes, respectively. A total of 29 differential lipids were detected between the control group and the whole brain irradiated group, among which 25 lipids were detected in the positive ion mode and 4 lipids were detected in the negative ion mode. The information on the 29 differential lipids detected in the untargeted lipidomics is listed in Appendix A. Regarding the distribution of non-modified lipids, we have listed the classification information of the 3031 total lipids detected in the Appendix A. The screening of differential metabolites was generally performed using a multiple test Fold Change (FC) > 1.5, a PLS-DA model (Appendix A) with a VIP value > 1, and a T test with a *p*-value < 0.1 by a volcano plot model (Appendix A). A heat map was created to display the changing degree of those altered lipids (Appendix A). Lipids are divided into eight categories according to the lipid classification system proposed by the International Lipid Classification and Nomenclature Committee funded by the Lipid Metabolites and Pathways Strategy (LIPID MAPS), including fatty acyls (FA), sterol lipids (SL), glycerolipids (GL), glycerophospholipids (GP), sphingolipids (SP), prenol lipids (PR), saccharolipids (SL), and polyketides (PK). As shown in Figure 2, all the significantly modified lipids belong to the following four categories, including GL, GP, SP, and FA. Diacylglycerol (DAG) and triacylglycerol (TAG) were classified as GL. Lysophosphatidylethanolamine (LPE), lysophosphatidylcholine (LPC), phosphatidylcholine (PC), and phosphatidylethanolamine (PE) were classified as GP. Sphingomyelin (SM), a type of SP, and branched fatty acid esters of hydroxy fatty acids (FAHFA) were also different between the two groups. As shown in Figure 2, the most significant differential lipids in the irradiated group were GPs and GLs.

As shown in Figure 2, DAG, TAG, LPE, SM, and FAHFA showed a decreasing trend upon radiation, with DAG as the most prominent differential lipid in the rat brain. The relative abundance of DAG in the irradiated group decreased by about 51% overall. In detail, 13 out of the 29 differential lipids were DAGs, including DAG 47:1, DAG 45:0, DAG 50:6e, DAG 47:0, DAG 46:0, DAG 53:1, DAG 49:2, DAG 47:7, DAG 48:0, DAG 53:0, DAG 47:8e, DAG 28:4, and DAG 42:11e. Among the differential DAGs, five down-regulated ones were saturated lipids and accounted for 38% of the differential DAGs. Eight unsaturated DAGs were down-regulated, except for DAG 28:4 and DAG 42:11e, which were up-regulated after heavy ion radiation. Compared to the control group, the DAG 46:0 decreased most significantly by about 80%. The relative abundance of DAG 47:1 was reduced by 67% after radiation. The remaining down-regulated DAGs changed from 45% to 60% (Figure 3A).

Similar to the down-regulation trend of DAG, a total of six differential TAGs were detected, including TAG 39:0, TAG 42: 5, TAG 43:0, TAG 44:7, TAG 45:6, and TAG 50:3. Among them, TAG 39:0, TAG 42: 5, TAG 43:0, and TAG 44:7 were decreased by approximately 70%. However, TAG 45:6 and TAG 50:3 exhibited significant post-irradiation increases. Compared to the control group, TAG 45:6 and TAG 50:3 were up-regulated by about 11% and 177%, respectively (Figure 3B).

As shown in Figure 3C, the irradiated PE content (PE 48:12e), PC (PC 32:0), and LPC (LPC 16:0) was up-regulated, but the content of SM (SM t36:1, SM d51:10), PA (phosphatidic acid, data not shown), and LPE (LPE 18:1) was down-regulated. In addition, the PE 48:12e, PC 32:0, and LPC 16:0 increased by more than 80%. Although the relative abundance of LPE decreased by 20% overall, LPE37 was raised by about 65%. Similarly, in the three significant SMs, SM t36:4 was up-regulated by 88%, but SM t36:1 and SM d51:10 were down-regulated by 74% and 60%, respectively. In summary, our results indicate that heavy ion radiation disturbs the central lipid metabolism.

### 2.4. Targeted Brain and Plasma Lipidomics

In order to confirm the differential lipids detected using the untargeted lipidomics method to support the hypothesized mechanism of brain damage, the differential DAGs and TAGs detected using the untargeted lipidomics method were quantified in the brain tissue and plasma samples using a targeted lipidomics platform. The brain and plasma lipid concentrations were statistically processed to identify intergroup variations. Overall, 11 out of 19 DAG and TAG differential lipids were detected and quantified in both the brain and plasma, probably due to the small number of samples. As shown in Table 2, lipid DAG 47:1, DAG 53:1, and DAG 50:6e in the rat brain tissue decreased by about 90%, 60%, and 60% compared to the control group, respectively (Figure 4A). The downward trends of these three lipids are consistent with their corresponding untargeted lipidomics. Among them, DAG 47:1 is significantly down-regulated in both targeted and untargeted lipidomics. However, it is interesting to note that the DAG 47: 1, DAG 53: 1, and DAG 50: 6e in the blood plasma of irradiated rats increased by 36%, 467%, and 195% compared to the plasma of the control rats, respectively (Figure 4B). Therefore, DAG 47:1, DAG 53:1, and DAG 50:6e in the rat brain tissue after the ^12^C^6+^ heavy ion irradiation had a negative correlation with plasma. However, the targeted quantitative lipidomics did not reveal any correlation for the other lipids.

### 2.5. Effects of Heavy Ion Irradiation on the Levels of Superoxide Dismutase (SOD) and Malondialdehyde (MDA) in the Rat Brain

Cellular water molecules have a greater chance of depositing energy and becoming ionized following heavy ion radiation exposure. Therefore, a series of primary radiolysis products will be produced by water molecule ionization, including free radicals and reactive oxygen species (ROS). Excessive ROS attack saturated or unsaturated fatty acids to initiate lipid peroxidation and produce a free radical chain reaction that eventually decomposes and produces toxic aldehydes [27]. As one of the main end products of lipid peroxidation, malondialdehyde (MDA) reflects the degree of lipid peroxidation in the body and indirectly reflects the degree of cell damage induced by excessive ROS. Superoxide dismutase (SOD) scavenges and converts harmful superoxide anion radicals into hydrogen peroxide. Hydrogen peroxide is broken down into water by catalase and peroxidase. We analyzed the central MDA content and SOD activity on the 7th day of whole-brain heavy ion radiation in the brain tissue (Figure 5). Although the SOD activity increased minutely after a week of heavy ion radiation, the changes were non-significant.

## 3. Discussion

While heavy ion radiation is widely used to treat cancer patients, it also induces more serious biological damage than X- or γ-ray radiation due to the sharp high-energy Bragg peaks [28,29]. Carbon ions have become a star beam for the effective treatment of brain tumors and melanomas due to their high LET, high precision, and low side effects [30]. Due to the advancement of system therapy and radiotherapy equipment technology in recent years, the survival period of radiotherapy patients is getting longer and longer. However, the late effects of radiation brain injury, including cognitive impairment, have enough time to manifest and seriously threaten the patient’s prognostic quality of life. Traditional radiotherapy is 1.5–2 Gy in daily fractions, and the total treatment lasts 5–6 weeks to reach a total dose of 50–70 Gy [31]. However, modern radiotherapy is rapidly changing the standard of cancer treatment. For example, extreme oligofractionation is a high-dose, single-fraction treatment [32]. The National Institute of Radiology (NIRS) of Japan uses a single delivered dose of a 25–30 Gy carbon ion beam to treat stage I/II non-small cell lung cancer [33]. Compared with fractional radiotherapy, a single high-dose radiotherapy has demonstrated a greater therapeutic advantage in limited clinical trials. However, damage to healthy tissues around the lesion and along the path of radiation is still a factor that needs to be considered during radiotherapy. Herein, we aim to find the potential biomarkers of radiation brain injury as early as possible, so that the early intervention and treatment of cognitive dysfunction that may occur in patients in the later period can be carried out.

The mid- and long-term effects of whole brain radiation are manifested as hippocampal-dependent functional defects, including the decline of space and object recognition memory [34]. Neuronal survival-rate reduction may be central to the pathogenesis of radiation-induced cognitive decline [35,36]. Moreover, heavy ion irradiation causes damage to the CNS, resulting in impaired cognitive performance, neuro-degeneration, and neuronal cell death [37]. In a long-term study, heavy ion irradiation persistently destroys the CNS and reduces animals’ behavioral performance [38]. However, it is not possible to see the impaired brain from the behavioral performance at the early post-irradiation stage.

A ^12^C^6+^ heavy ion beam with a total of 15 Gy (260 MeV, LET = 13.9 KeV/μm) low LET plateau irradiation was used to irradiate the whole rat brain. In order to detect changes before behavior differences could be observed, a lipidomic analysis identified and analyzed lipids of differential expression. In 2018, Hinzman et al., did not observe lipid changes in the hippocampus after two days of cranial X-ray irradiation [39]. However, lipid metabolic disorders were found after two weeks of irradiation [39]. Based on previous findings, behavioral changes should occur in the early-delayed and late stages post-irradiation [40]. Therefore, the acute phase (7d) after irradiation was selected in our study. To the best of our knowledge, this is the first published study that described the effects of heavy ion irradiation on the lipidomic profiles at the whole brain level in rats using UPLC–MS technology.

Lipids are a large class of compounds that represent essential structural components in the brain. They play an irreplaceable role in neural function, membrane formation, neurite growth, and signal transduction [41]. Our aim is to reveal the changes in lipid levels in the acute phase of brain injury caused by whole brain carbon heavy ion radiation. Lipidomic detection in the brain helps us understand the changes in lipids directly related to radiation brain damage and how lipids affect the brain structure and function. Lipidomic detection in plasma can be used as a potential biomarker for radiation brain damage.

We first conducted untargeted lipidomic analyses to build up knowledge of differential lipids. Among the 3031 lipids detected by UPLC–MS, phosphatidylserine (PS), phosphatidylinositol (PI), and cholesteryl ester (CE) were detected but not significantly dysregulated upon heavy ion radiation. GPs and GLs were the most preferentially modified lipids. GPs are key components of the membrane structure, stability, fluidity, and permeability, and are essential for membrane proteins, receptors, and ion-channel function [42,43,44]. DAGs are important precursors to the formation of various phospholipids, and participate in the metabolic cycle of lipids and hormones as an intermediate product of lipid metabolism [45,46]. In addition, DAGs act as a secondary messenger, phosphorylating a range of substrate proteins by activating protein kinase C (PKC) [47]. On the other hand, TAGs provide a major source of energy and constitute a critical component of the lipoprotein [48]. Our untargeted lipidomic studies revealed that the contents of DAGs and TAGs in the irradiated group were significantly down-regulated compared to the controls, indicating that differential lipids might be dysregulated upon irradiation (Figure 2).

DAGs can be phosphorylated to phosphatidic acid (PA) by diglyceride kinase (DGK) to further regulate cell function [49]. There are five subtypes of DGK, among which DGKβ is expressed in neuronal cells and plays an important role in brain function [50,51]. Studies have shown that DGKβ knockout mice exhibit cognitive impairment and anxiety [52,53]. TAGs can promote the transport of peripherally derived peptides ghrelin and insulin that have a positive effect on cognition in the blood–brain barrier [54]. Considering the functional relevance of DAGs and TAGs and based on the results of untargeted lipidomics, we conducted a targeted quantitative analysis of 19 DAGs and TAGs that showed differences (Figure 3A,B) in both brain tissue and plasma. Overall, 11 out of 19 DAG and TAG differential lipids were detected and quantified in both the brain and plasma, probably due to the small number of samples. Combined with the changes in the relative abundance of each DAG and TAG lipid in the control group and the radiation group in Figure 4, we found that the contents of DAG 47:1, DAG 53:1, and DAG 50:6e in the rat brain tissue negatively correlated with those in the plasma, possibly due to the release of DAG 47: 1, DAG 53: 1, and DAG 50: 6e from the damaged brain tissue to the blood plasma. Among them, DAG 47:1 in the brain tissue post irradiation showed significant down-regulation in both untargeted and targeted lipidomics, suggesting that DAG 47:1 may be a lipid directly related to radiation brain damage. The results of targeted lipidomics in plasma showed that the relative content of DAG 47:1 in the irradiated group had an up-regulation trend, indicating that DAG 47:1 could be used as a potential biomarker of radiation brain damage. However, due to the small sample size in this experiment, the error bar is too large, so no significant trend of change is seen.

Radiation ionizes molecules, inducing chemical reactions. The highly reactive chemical species generated by radiation through water radiolysis can chemically discriminate other molecules [55]. The high energy protons generated by ionizing radiation leads to the ejection of electrons from water molecules via hydrolysis in biological systems where water is abundant. In the presence of oxygen, free electrons interact with oxygen and form superoxide radicals (O_2_·^−^) [56]. O_2_ generates hydrogen peroxide (H_2_O_2_) via enzymatic or spontaneous dismutation. H_2_O_2_ in turn generates hydroxyl radicals (·OH) via the Fenton reaction or the Haber–Weiss reaction. Heavy ion radiation has been reported to trigger intracellular ROS generation by activating nicotinamide adenine dinucleotide phosphate (NADPH) oxidase in membranes and subsequently causing the dysfunction of mitochondria [57]. Malondialdehyde (MDA) is formed as a product of either non-enzymatic lipid oxidation or enzymatic arachidonic acid oxygenation in the cyclooxygenase pathway [58]. Free radicals generate the lipid peroxidation process in an organism. MDA is one of the final products of polyunsaturated fatty acid peroxidation in the cells. An increase in free radicals causes the overproduction of MDA. It has been used as a marker to indicate the amount of ROS and the level of lipid peroxidation. SOD, an enzyme which catalyzes the dismutation of the superoxide anion radical (O_2_·^−^) into molecular oxygen (O_2_) and hydrogen peroxide (H_2_O_2_), is another indicator of the redox level in the body. The MDA content in rat brain tissue is significantly increased and the SOD activity was significantly reduced after exposure to systemic or cranial irradiation compared to the control group [59]. This suggests that lipid peroxidation plays an important role in the acute phase of irradiation. However, the lipid peroxidation decreased with time. Our results show that there were neither significant differences in the MDA content nor the SOD activity between the irradiated- and the sham group. We hypothesized that this was due to the self-healing ability to balance the excessive ROS within a week after heavy ion irradiation.

Studies have shown that radiation damages the CNS microvasculature and causes microglial activation endoplasmic reticulum (ER) stress [60,61,62]. The endoplasmic reticulum is the largest multi-functional organelle which plays an indispensable role in lipid synthesis, transport, and degradation in cells [63]. It is the synthetic site for a large number of structural phospholipids, sterols, and storage lipids, such as TAGs and sterol esters. In addition, PC is the main phospholipid of the endoplasmic reticulum membrane [64]. ER stress is a protective response of eukaryotic cells and refers to a subcellular pathological state in which the physiological function of the ER is disturbed upon stress. Studies have shown that excessive or persistent ER stress can induce neuronal apoptosis through various signaling pathways such as protein kinase RNA-like endoplasmic reticulum kinase (PERK)/ C/EBP-homologous protein (CHOP) in neurological diseases such as cerebral ischemia-reperfusion and Alzheimer’s disease [65,66]. Specifically, extracellular Aβ and intracellular ER are connected through calcium (Ca^2+^) [67]. Excessive or long-lasting ER stress can lead to the excessive accumulation of folded proteins, Aβ conformation changes and aggregation, and protease hydrolysis being difficult, resulting in neurotoxic effects [68]. At the same time, a significant increase in phosphorylated tau protein will be found [69]. Excessive or persistent ER stress eventually leads to cell damage and cognitive decline. Interestingly, the majority of the differentially modified lipids such as DAGs we observed have been reported to be involved in ER-related catalytic pathways [70]. DAG composition can be altered through phospholipid hydrolization, such as PCs and Pes, by phospholipase C (PLC) [71]. They can also be hydrolyzed by PAs through the corresponding enzyme and by TAGs through lipase. Widely abundant in the brain, Phospholipase D (PLD) hydrolyzes PCs to form PAs [72]. Phospholipase A2 (PLA2) hydrolyzes phospholipids to form fatty acid and LPL products, which are important mediators of phospholipase remodeling [73,74]. We hypothesized that PLD, PLC, and PLA2 may all be in a hypoactive state due to their disturbed metabolic pathways after heavy ion irradiation. In the results of an untargeted lipidomic analysis, we observed the accumulation of PC and PE and the reduction of PA, DAG, and LPE. This indicates that heavy ion irradiation exposure may impose ER stress in the brain (Figure 6). This hypothesis will be verified in our follow-up work.

## 4. Materials and Methods

### 4.1. Animal Housing and Irradiation

The 7-week-old male Wistar wild-type rats were purchased from Lanzhou University Medical Center and Animal Center. Upon arrival, the rats were maintained in sterile housing conditions at 25 °C, with a 12 h light-dark schedule and ad libitum access to food and water. The rats were fed with standard maintenance feed from Beijing Keao Xieli Feed Co., Ltd. The rats were randomly divided into a control (*n* = 8) or irradiation group (n=8). The rats in each group underwent five days of adaptive feeding before irradiation. The rats in the irradiation group were anesthetized with an intraperitoneal injection of pentobarbital sodium (60 mg/kg) and a single whole head ^12^C^6+^ irradiation at a dose rate of 0.5 Gy/min for a total dose of 15 Gy (260 MeV, LET = 13.9 KeV/μm) at Wuwei Heavy Ion Hospital. The control animals were also anesthetized but did not receive any irradiation. Upon radiation exposure, the rat’s head was aligned with the radiation beam window. No special protective measures were taken for other parts of the body. All the procedures were performed in accordance with the ethics guideline and approved by Beijing Institute of Technology (SYXK-BIT-20190624003). All efforts were made to minimize animal suffering and to use as few animals as necessary.

### 4.2. Hematoxylin and Eosin Staining

On the 7th day post-irradiation, the rats were decapitated. One rat from each group was randomly selected, and the left hemisphere was fixed in 4% paraformaldehyde fixative solution for 4–6 h. After fixation, the tissues were dehydrated and embedded in paraffin. The 4 μm brain tissue sections were cut and air-dried on glass slides. After the paraffin sections were dewaxed, hematoxylin staining was performed for 3 min, followed by eosin staining for 3 min. The sections were then dehydrated with alcohol, made hyaline with xylene, and sealed. The cell morphology of the hippocampus and cortex were observed under the light microscope. The denatured cells are defined by their shrunken, pyknotic morphology and eosinophilic staining.

### 4.3. Brain Tissue and Blood Plasma Preparation and Lipid Extraction

Seven days after irradiation, the rats were anesthetized with 60 mg/kg of pentobarbital sodium intraperitoneal injection. An amount of 2 mL of blood was taken from the inferior vena cava of the rat to a centrifuge tube filled with EDTA-K_2_ in advance. After centrifugation at 3000 rpm for 10 min, the supernatant was taken and stored at −80 °C. After quickly excising the brain, the brain tissue was frozen in liquid nitrogen before storage at −80 °C until use. Due to limited testing costs, a random three samples in each group were chosen for lipidomic analysis.

The whole brain total lipids were extracted with methyl-tert-butyl ether (MTBE). After whole brain homogenization, pre-cold methanol (300 μL) was added to the tissue homogenate (65 mg), and then MTBE (1000 μL) was added. The resultant mixture was vortexed for 1 h, followed by the addition of 300 µL of water and vortexing for another 5 min. After mixing vigorously and centrifugation at 12,000 rpm for 10 min at 4 °C, the upper organic phase was carefully collected (approximately 400 μL). The 400 µL of supernatant was re-dissolved in 200 µL of acetonitrile/isopropanol (*v/v*, 1:1) and centrifuged at 12,000 rpm for 10 min. We took 100 μL of the supernatant solution and placed in a 250 μL liquid-lined pipe to be tested.

After each plasma sample was shaken and mixed, 50 μL was taken in a 1 mL centrifuge tube, followed by 300 μL of dichloromethane/methanol (*v/v*, 3:1). The sample was shaken for 1 h, and 100 μL of double-distilled water was added to the centrifuge tube to mix. The sample was then centrifuged at 12,000 rpm for 10 min, and 150 μL of the dichloromethane layer (lower layer) was taken to the labeled sample tube to concentrate and dry. Finally, the sample was re-dissolved with 100 μL of isopropanol/acetonitrile (*v/v*, 1:1) and centrifuged at 12,000 rpm for 10 min. A volume of 80 μL of the supernatant solution was placed in a 250 μL liquid-lined pipe to be tested.

### 4.4. Mass Spectrometry Analysis of Lipid Metabolites

For the analysis of each sample, lipid samples were injected by a Nexera UPLC LC-30A (SHIMADZU; UPLC system with ACQUITY UPLC C8 Column (1.7 µm, 2.1 × 100 mm; Waters, MA, USA)) and analyzed by TripleTOF5600+ (AB SCIEX™). The injection volume was fixed at 1 μL. The C8 column was maintained at 40 °C. The flow rate of the mobile phase was 0.3 mL/min. Mobile phase A consisted of acetonitrile/water = 6/4 (5 mM ammonium acetate). Mobile phase B consisted of acetonitrile/isopropanol = 1/9 (5 mM ammonium acetate). A linear gradient was used as follows: 20 % B at 0–1 min, 20–100 % B at 1–13 min, 20 % B at 13–16 min.

The mass spectrometry (MS) was separately operated in the positive and negative modes. The scan range of the instrument was set at a mass-to-charge ratio (*m/z*) of 50–1500. The positive and negative ion source spray voltages were 3.7 kV and 3.5 kV, respectively. The capillary heating temperature was 320 °C. The sheath gas and auxiliary gas pressure were 30 psi and 10 psi, respectively. The internal standard 20:0–20:1–20:0 D5 TG (1,3(d5)-dieicosanoyl-2-(11*Z*-eicosenoyl)-glycerol), purchased from Sigma-Aldrich (Shanghai, China) Trading Co., Ltd., was used to quantify the targeted lipids.

### 4.5. Data Processing and Analysis

The UPLC–ESI–TOF–MS data was processed by MS–DIAL 3.96 data analysis software [75]. The analysis process was sequentially performed from the alignment and peak picking to the identification of lipids. The metabolites were identified by mainly referring to the Lipid Maps Database (www.lipidmaps.org). All the identified lipid molecules are expressed in relative quantitation (i.e., peak area). The data were processed by an unsupervised principal component analysis (PCA), supervised with multivariate regression techniques (PLS–DA), and supervised by orthogonal partial least-squares discriminant analysis (OPLS–DA) methods to obtain group clusters. PCA is an unsupervised data analysis tool used to reduce and visualize multidimensional data (X) without referring to class labels (Y). The data were summarized into fewer variables called principal components, and the weighted average of the original variables was found. OPLS–DA is often used in lieu of PLS-DA to disentangle group predictive and group-unrelated variation in the measured data. The OPLS-DA model quality is assessed by its ability to fit (R^2^) and predict (Q^2^) the variance of the data. R^2^X and R^2^Y respectively represent the interpretation rate of the metabolite (X) and grouping situation (Y) of the model, and Q^2^ represents the accuracy rate of the model prediction. To estimate the significance of Q^2^ and R^2^, a permutation test with 100 permutations was used. In doing so, OPLS-DA constructs more parsimonious and easily interpretable models compared to PLS-DA. The selection of potential biomarkers is generally performed by the fold change (FC) > 1.5, the VIP value of the PLS-DA model > 1, and the *p*-value of the T-test < 0.05.

### 4.6. Measure of Oxidative Stress

Malondialdehyde (MDA), a lipid hydroperoxide degradation end product, forms a red product with thiobarbituric acid (TBAR) at 90–100 °C, which has a maximum absorption peak at 532 nm. The levels of MDA in the brain tissues were measured using the TBA Assay Kit (Nanjing Jiancheng Bioengineering Institute). WST-1, which is 2-(4-Iodophenyl)-3-(4-nitrophenyl)-5-(2,4-disulfophenyl)-2H-tetrazolium monosodium salt, reacts with the superoxide anion catalyzed by xanthine oxidase to form a water-soluble formazan. This reaction step can be inhibited by SOD. In this experiment, the activity of SOD enzyme in the brain tissues was determined by the WST-1 Assay kit (Nanjing Jiancheng Bioengineering Institute). Both assays were carried out in triplicate in accordance with the kit instructions. Both the results were corrected for the protein level.

### 4.7. Statistical Analysis

Data were expressed as mean ± standard deviation (SD). Differences between the groups were examined for statistical significance using unpaired, two-tailed Student’s t-tests. A *p* < 0.05 was considered statistically significant.

## 5. Conclusions

In summary, to shed some light on the early stage lipid molecular damages caused by heavy ion irradiation before different behavioral performance could be observed, untargeted and targeted UPLC-MS lipidomic analyses were conducted to detect potential differential lipid biomarkers on the whole brain and blood plasma levels of rats exposed to the 15 Gy ^12^C^6+^ ions plateau irradiation on the whole head. It was found that heavy ion irradiation induced a profound alteration in the lipidomic file in the rat brain. The predominantly modified lipid species include GPs and GLs, with DAG 47:1 as the most significant differential lipids. Targeted lipidomics proved that the content of DAG 47:1 in the rat brain tissues decreased after irradiation, and the change trends were opposite to those of the blood plasma. This obvious differential amount of DAG 47:1 in the rat brain tissues and blood plasma after irradiation indicate that they can be potentially used as biomarkers of cognitive dysfunction caused by heavy ion ^12^C^6+^ irradiation. In addition, we proposed that the lipid metabolism disorder induced by heavy ion irradiation might be caused by the endoplasmic reticulum stress. The availability of lipid profiles will expand our understanding of the consequences of heavy ion irradiation and lay the foundation for the subsequent exploration of the molecular mechanisms of radiation brain damage.

## Figures and Tables

**Figure 1 molecules-25-03762-f001:**
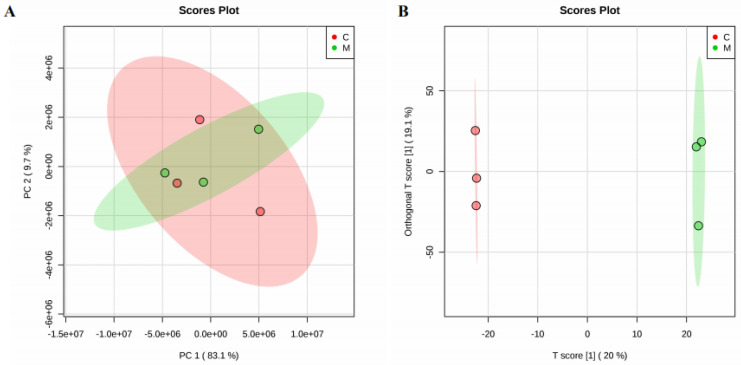
PCA (**A**) and OPLS-DA (**B**) plot visualizing the lipidomic profiles of the brain tissue (*n* = 3) derived from sham-irradiated rats (C red) vs. 15 Gy whole-brain irradiated rats (M green) after 7 days.

**Figure 2 molecules-25-03762-f002:**
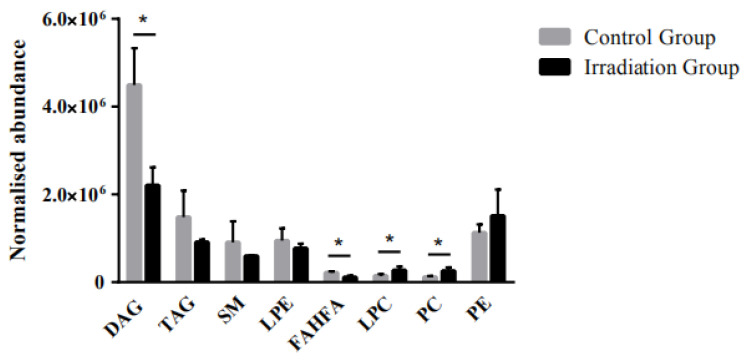
Representative overview of validated classes of lipids that were dysregulated in the brain tissue of rats receiving 15 Gy low low linear energy (LET) plateau whole-brain irradiation compared to the controls. The relative levels of lipid species were expressed with error bars as SD. Lipid class data were conducted based on the unpaired and two-tailed Student’s t-test (* *p* < 0.05, *p* < 0.1 without *, *n* = 3).

**Figure 3 molecules-25-03762-f003:**
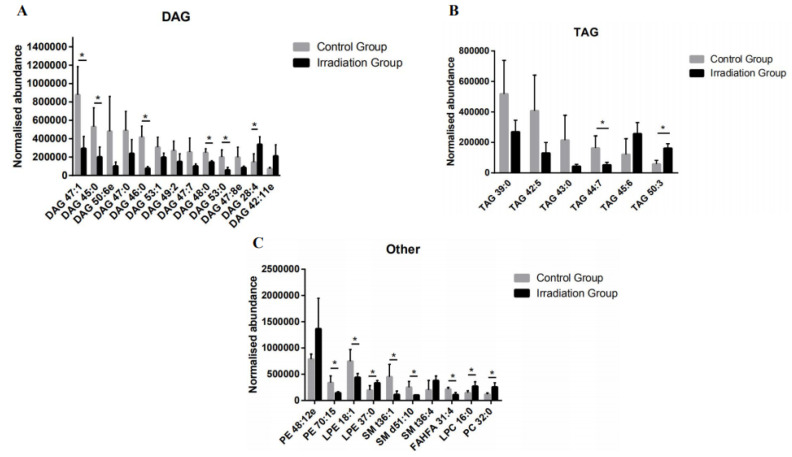
Diacylglycerol (DAG) (**A**), triacylglycerol (TAG) (**B**), and other (**C**) detailed lipids that were dysregulated in the rat brain tissue receiving 15 Gy low LET plateau whole-brain irradiation compared to the sham-irradiated controls. The relative levels of individual lipid species were expressed with error bars as SD. Lipid class data were conducted based on the unpaired and two-tailed Student’s t-test (* *p* < 0.05, *p* < 0.1 without *, *n* = 3). “e” means ether.

**Figure 4 molecules-25-03762-f004:**
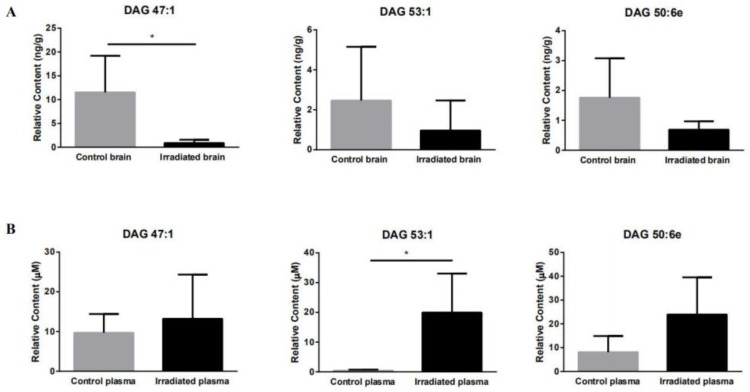
Quantitative analysis of DAG 47:1, DAG 53:1, and DAG 50:6e in the rat brain tissue (**A**) and plasma (**B**) after 15 Gy low LET plateau whole-brain irradiation. (**A**) Quantification of DAG 47:1, DAG 53:1, and DAG 50:6e in the rat brain. (**B**) Quantification of DAG 47:1, DAG 53:1, and DAG 50:6e in the rat plasma. The relative levels of individual lipid species were expressed with error bars as SD. Lipid class data were found with the unpaired and two-tailed Student’s t-test (* *p* < 0.05, *p* < 0.1 without *, *n* = 3).

**Figure 5 molecules-25-03762-f005:**
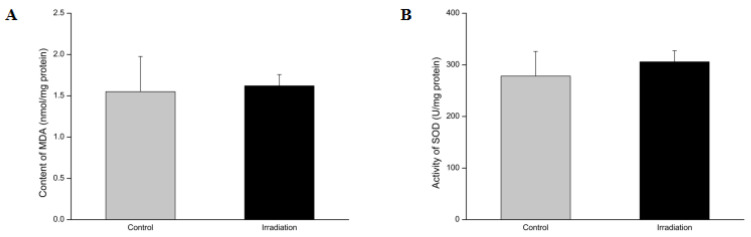
Effects of 15 Gy low LET plateau whole-brain irradiation heavy ion irradiation on the malondialdehyde (MDA) content (**A**) and superoxide dismutase (SOD) activity (**B**) in rat brain tissue (*n* = 3) 7 days post-irradiation. The relative levels are expressed with error bars as SD.

**Figure 6 molecules-25-03762-f006:**
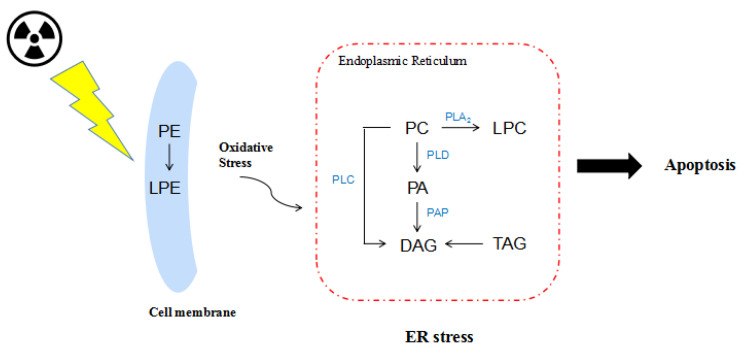
Hypothesized damage of glycerophospholipid metabolism under heavy ion irradiation. Glycerophospholipid metabolism is impaired under heavy ion irradiation, with concerted phosphatidylcholine (PC) increase downregulating its metabolites phosphatidic acid (PA) and DAG, possibly by inhibited PLD, PAP, and PLC enzyme activity, respectively (red). PLA2, phospholipase A2; PLC, phospholipase C; PLD, phospholipase D; PAP, phosphatidate phosphatase.

**Table 1 molecules-25-03762-t001:** Brain weight difference between the control and irradiated rats. Expressed as mean ± SD. (* *p* < 0.05, *n* = 3).

	Sample	Brain Weight (g)		Sample	Brain Weight (g)
Control Group	C1	1.737	Irradiation Group	M1	1.566
	C2	1.777		M2	1.515
	C3	1.686		M3	1.507
Average ± SD		1.733 ± 0.045	Average ± SD		1.529 ± 0.032 *

**Table 2 molecules-25-03762-t002:** Quantitatively annotated DAG and TAG species dysregulated 7 days after 15 Gy low LET plateau whole-brain irradiation. Expressed as mean ± SD, *n* = 3.

Average Rt (min)	Average Mz	Metabolite Name	Concentration in Control Brain (ng/g)	Concentration in Irradiated Brain (ng/g)	Concentration in Control Plasma (μM)	Concentration in Irradiated Plasma (μM)
14.368	766.653	DAG 47:8e	0.424 ± 0.304	2.368 ± 2.704	3.500 ± 3.658	3.357 ± 0.463
14.381	768.749	DAG 45:0	1.517 ± 1.209	6.011 ± 5.384	8.241 ± 7.510	8.301 ± 11.660
14.611	782.654	DAG 47:7	0.148 ± 0.064	4.080 ± 4.339	2.028 ± 0.580	2.120 ± 1.886
15.231	794.750	DAG 47:1	11.522 ± 7.706	0.889 ± 0.676	9.687 ± 4.733	13.210 ± 11.129
13.861	810.792	DAG 48:0	0.729 ± 0.280	2.419 ± 2.175	5.891 ± 6.456	7.446 ± 6.937
14.346	812.743	DAG 50:6e	1.758 ± 1.321	0.690 ± 0.277	8.104 ± 6.811	23.919 ± 15.628
14.685	820.794	DAG 49:2	0.386 ± 0.315	4.697 ± 5.265	1.471 ± 1.157	2.823 ± 2.499
14.776	878.824	DAG 53:1	2.465 ± 2.686	0.964 ± 1.502	0.418 ± 0.316	19.943 ± 13.097
9.466	735.552	TAG 42:5	0.070 ± 0.055	7.222 9.748	0.409 ± 0.367	0.272 ± 0.106
14.633	759.618	TAG 43:0	0.163 ± 0.141	3.132 ± 2.693	2.088 ± 1.894	1.632 ± 0.253
14.706	851.689	TAG 50:3	1.204 ± 0.800	0.023 ± 0.010	4.127 ± 4.254	5.796 ± 4.169

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
