# Peer review of "Effect of Heavy Ion 12C6+ Radiation on Lipid Constitution in the Rat Brain"

_molecules, 2020, doi:10.3390/molecules25163762_

Round 1
Reviewer 1 Report
The major focus of the manuscript presented by Bo Li et al. was set on the effect of Heavy Ion 12C6+ on lipid constitution in the rat brain.
Overall the topic of the manuscript is interesting but the study lacks relevant justifications in many parts e.g. time of euthanasia and delivered dose, final aim and experimental choices are difficult to understand. Moreover, target and non-targeted lipidomics bring different results without any clear explanations; for instance only two lipids are statistically significant after target lipidomics but discussion is based on the full non targeted results. This paper mainly comments bibliography and focuses on the danger of heavy ion radiation but few comments and critical analysis are made on its own results, acquired with a huge acute dose of 12C6+. Major and minor points of criticism are given successively.
Minor points:
- English needs to be corrected in some parts : gamma not gama (l33); heavy not heave (l100); food not chow (l396), etc
- L407 - Hematoxylin and Eosin Staining: Figure 1: scale bars are missing in the pictures. Only tiny sections of cortex and hippocampus are presented. What about the gyrus dentatus and other CA parts from hippocampus for instance? What is the number of replicates for these analyses?
- In Table 1, please check the significant figures. Same comment for Table 2.
- Add in all captions the number of replicates used for each analysis.
- l124: explain why you mention the “colon” in this section
- OPLS-DA analysis needs to be better describe in experimental section.
- Through the whole document what does the “e” mean at the end of some lipid names eg 50:6e?
Major points:
- Dose justification is really unclear; is it really realistic? Or is it better to simply decorrelate this assay from biological significancy? From many papers, space doses (cumulated) are around 0,6mGy per day, during 6 months for instance (ISS periods), it leads to a dose around 1,1 Gy (whole body, chronic dose); parallely for hadrontherapy, dose delivered are ~1,5-2Gy (acute). The choice of 15Gy is then difficult to understand for rodent, above all when authors compare this paper with another one, with a 9Gy irradiation.
- In the same way, what is the justification of time post exposure for euthanasia excepted that it is one week more than the previous study? Is there any physiological relevance for this choice?
- Finally, what happened with the other 10 animals (5 control, 5exposed)? It is really difficult to understand. Did they die at three month years old?
- Table 1 and S1: It can be understandable to have only 3 replicates for lipidomics because of prices (and still, it is really not enough) but why body weights, histology, etc have not been done on the 8 replicates?
- L227 - Targeted Brain and Plasma Lipidomics: Among all lipids measured only two appeared to be significantly modulated in brain or in plasma; how do you explain the differences with the non-targeted lipidomics? It would be probably more relevant to build the discussion on these latter results. Error bars are very large; what is the number of analyses for this analysis?
Author Response
Minor points:
- English needs to be corrected in some parts : gamma not gama (L33); heavy not heave (L100); food not chow (L396), etc
Response:We thoroughly revised the manuscript to improve the content and English quality. Our collaborator Prof. Nafissa Ismail at the University of Ottawa, Canada, contributed to the discussion and manuscript revision.
- L407 - Hematoxylin and Eosin Staining: Figure 1: scale bars are missing in the pictures. Only tiny sections of cortex and hippocampus are presented. What about the gyrus dentatus and other CA parts from hippocampus for instance? What is the number of replicates for these analyses?
Response:We revised Figure 1, the results, and discussion with two neurobiologists who have experience in brain research. We added a full and zoomed-in images of the prefrontal cortex, hippocampus CA, and DG area in Figure 1, increased the magnification of the pictures, and displayed the scale bar value under each picture. Based on our previous study we did not expect obvious damage. Therefore, we only did analysis on one sample. Please see below for details in the manuscript highlighted from L112 to L132 in section 2.2.
Radiation-induced cognitive dysfunction frequently occurs in the hippocampus and prefrontal cortex [27, 28]. In order to determine the effects of 12C6+ irradiation on the rat brain, we randomly selected a rat's left hemisphere from each group to make paraffin sections and H&E staining analysis. Figure 1A and Figure 1B represent the general views of the left hemispheres of rats in the control and irradiation groups, respectively (bar =1000 μm). Figure A1 and Figure B1 represent the general view of the prefrontal cortex of rats in the control group and the irradiated group after magnification of 3 times (bar = 500 μm). The cell morphology of the prefrontal cortex of rats in the control group and the irradiated group after 40 times magnification (bar = 20 μm) was checked as shown in Figure A4 and Figure B4. Compared with the control group, the neuron nuclei in the prefrontal cortex of the irradiated rats shows vacuolation (black arrow in Figure B4), but no inflammatory infiltration can be observed. Pathological analysis of the hippocampus, another sensitive area of brain radiation damage was also conducted. Figure A2 and Figure A3 represent the general view of the cornu ammonis (CA) area and dentate gyrus (DG) area of the hippocampus of the control group, respectively (bar = 100 μm). Figure A5 and Figure A6 are the partial views of the CA area and DG area after 40 times magnification (bar = 20 μm). Compared with Figure B2 and Figure B3 (bar = 100 μm), which also represent the CA area and DG area of the irradiated group, the neurons in the hippocampus are abundant and closely arranged in both control and irradiated group. The neuron morphology is normal and the nucleolus is obvious. Figure B5 and Figure B6 can be seen in enlarged views of the CA and DG areas of the hippocampus in the irradiated group (bar = 20 μm). The results of the above pathological analysis show that the acute period after 15 Gy 12C6+ irradiation did cause morphology changes on cells, although not obvious in the rat brain after 7 days.
- In Table 1, please check the significant figures. Same comment for Table 2.
Response:We revised the significant figures in both Table 1 and Table 2 to include only three significant figures.
- Add in all captions the number of replicates used for each analysis.
Response:We added the number of replicates for each analysis.
- L124: explain why you mention the “colon” in this section
Response:It was a typing error and we revised it.
- OPLS-DA analysis needs to be better describe in experimental section.
Response: We have revised the manuscript according to the reviewer’s suggestion. Please see below for details in the manuscript highlighted in section 2.3 and section 4.5.
“OPLS-DA is an extended form of partial least squares-discriminant analysis (PLS-DA). In addition to explaining overall differences between classes, OPLS-DA separates predictive and non-predictive variation.”
“The OPLS-DA model quality is assessed by its ability to fit (R2) and predict (Q2) variance of the data. R2X and R2Y respectively represent the interpretation rate of the metabolite (X) and grouping situation (Y) of the model, and Q2 represents the accuracy rate of model prediction. To estimate the significance of Q2 and R2, permutation test with 100 permutations is used.”
- Through the whole document what does the “e” mean at the end of some lipid names eg 50:6e?
Response:“e” means ether. DAG 50:6e is ether DAG, which also belongs to DAG. We added the explanation when it first appeared according to the reviewer’s suggestion. Please see the revised figure legend for Figure 4 highlighted in the manuscript.
Major points:
- Dose justification is really unclear; is it really realistic? Or is it better to simply decorrelate this assay from biological significancy? From many papers, space doses (cumulated) are around 0,6mGy per day, during 6 months for instance (ISS periods), it leads to a dose around 1,1 Gy (whole body, chronic dose); parallely for hadrontherapy, dose delivered are ~1,5-2Gy (acute). The choice of 15Gy is then difficult to understand for rodent, above all when authors compare this paper with another one, with a 9Gy irradiation.
Response:We selected low LET plateau irradiation with a small deposition dose (260 MeV, LET = 13.9 KeV/μm). The plateau region of heavy ion beam and X-rays belong to low linear energy (LET) radiation. The radiation biological effect produced by the plateau region of heavy ion beam is much lower than the radiation biological effect produced in or near the Bragg peak area. Again, anything that could happen in high doses could also theoretically occur at low doses, but we may have to introduce high doses to see the effects more clearly, resulting in more-accurate biomarker inventions.
Our team’s previous study (Zhang et al., 2019) on the sensitivity of rat brain regions under gamma radiation showed that the weight of rats dropped significantly on the 7th day after a single dose of 30 Gy of whole brain gamma irradiation, and then began to gradually recover, and the mortality of rats after irradiation is zero. At the same time, it was observed that the immune function and hematopoietic function of the irradiated rats were acutely damaged, which was manifested as a significant decrease in the content of white blood cells and platelets in the peripheral blood. Based on our research experience, it was suggested that the biological effects of radiation caused by carbon heavy ion irradiation in the plateau at the same dose are about twice that of gamma irradiation. In addition, we previously constructed the rat model of brain-localized 15 Gy carbon ion radiation (plateau region) and the rats can survive for more than three months after irradiation (Lei et al., 2015). Based on the above research experience, we continued to choose the dose and time point 7 days (acute phase) after 15 Gy whole brain irradiation with carbon heavy ions in the plateau as the irradiation conditions for this experiment.
According to the reviewer’s suggestion and after discussions with researchers in the field of radiation biology, we revised the background of the manuscript to avoid ambiguity. Please see below for details in the manuscript highlighted in Introduction and Discussion.
Introduction: In biomedical research, heavy ion beam currents are used to investigate the biological effects of ground-based simulated cosmic rays. They are also commonly used in radiotherapy to treat cancer patients [1]. Unlike X-ray and gamma radiation, heavy ions have a reverse depth dose distribution and high relative biological effect (RBE), making them ideal tools for head, neck, and central nervous system tumors [2]. Furthermore, cranial radiotherapy remains the mainstay of palliative treatment in patients with multiple symptomatic brain metastases [3]. Radiotherapy does not only target the tumors, but also affects the healthy brain tissue surrounding the tumor. However, research on the effect of radiotherapy on healthy brain tissue is limited. Studies have shown that heavy ion radiation causes structural and functional damage in the CNS regardless of low or high LET irradiation [4, 5]. Clinically, CNS damage caused by whole brain irradiation is divided into acute (≤7d), early-(up to months) and late-delayed stages (several months to years) [6]. Brain histopathology at the late-delayed stage following medium or high irradiation with heavy ions showed irreversible cerebral spinal cord disorders including vascular injury and white matter necrosis. Behavioral consequences of spinal cord disorders include cognitive dysfunction and abnormal walking patterns [7, 8]. In contrast, the pathological and behavioral consequences of the acute stage following irradiation are more subtle [8]. Non-invasive, early onset, biomarkers have been developed in response to heavy-ion radiation damage mediated cognitive decline.
Discussion: While heavy ion radiation is widely used to treat cancer patients, it also induces more serious biological damage than X- or gamma-ray radiation due to the sharp high-energy Bragg peaks [28, 29]. Carbon ions have become a star beam for effective treatment of brain tumors and melanomas due to their high LET, high precision and low side effects [30]. Due to the advancement of system therapy and radiotherapy equipment technology in recent years, the survival period of radiotherapy patients is getting longer and longer. However, the late effects of radiation brain injury, including cognitive impairment, have enough time to manifest and seriously threaten the patient’s prognostic quality of life. Traditional radiotherapy is 1.5-2 Gy in daily fractions, and the total treatment lasts 5-6 weeks to reach a total dose of 50-70 Gy [31]. However, modern radiotherapy is rapidly changing the standard of cancer treatment. For example, extreme oligofractionation is high-dose, single-fraction treatment [32]. The National Institute of Radiology (NIRS) of Japan uses a single delivered dose of 25-30 Gy carbon ion beam to treat stage I/II non-small cell lung cancer [33]. Compared with fractional radiotherapy, a single high-dose radiotherapy has demonstrated a greater therapeutic advantage in limited clinical trials. However, damage to healthy tissues around the lesion and along the path of radiation is still a factor that needs to be considered during radiotherapy. Herein, we aim to find potential biomarkers of radiation brain injury can be discovered as early as possible, so that early intervention and treatment of cognitive dysfunction that may occur in patients in the later period can be carried out.
- In the same way, what is the justification of time post exposure for euthanasia excepted that it is one week more than the previous study? Is there any physiological relevance for this choice?
Response: We have revised the manuscript with more explanation. Please see below for details in the manuscript highlighted in Introduction.
“Clinically, CNS damage caused by whole brain irradiation is divided into acute (≤7d), early-(up to months) and late-delayed stages (several months to years) (Belka et al., 2001).
Because the brain is rich in lipids, we hope to explore the mechanism of radiation-induced damage to the central nervous system from the perspective of lipids. In order to detect changes before behavior differences could be observed, lipidomic analysis identified and analyzed lipids of differential expression. To the best of our knowledge, this is the first published study that described the effects of heavy ion irradiation on the lipidomic profiles at the whole brain level in rats using UPLC-MS technology.
- Finally, what happened with the other 10 animals (5 control, 5exposed)? It is really difficult to understand. Did they die at three month years old?
Response: The high cost of lipidomics restricted the sample size we were economically capable of analyzing, we randomly selected 3 rats from each group to analyze lipidomics. Although we had cryogenically preserved our remaining samples for later analysis, we lost them due to a university shutdown following COVID-19.We decided to submit the results as they are because we believe they are valid, reliable and important in their current form. Although the number of samples is limited we believe the robust effects are of high quality. According to the reviewer’s suggestion, we supplemented the weight data of the remaining rats in Table S1 in the “supporting information”.
- Table 1 and S1: It can be understandable to have only 3 replicates for lipidomics because of prices (and still, it is really not enough) but why body weights, histology, etc have not been done on the 8 replicates?
Response: According to the reviewer’s suggestion, we supplemented the weight data of the remaining rats in Table S1 of the supporting information. For histopathological analysis, we only randomly selected the left hemisphere of one of each group for paraffin sectioning and H&E staining. Based on our previous study, we did not expect obvious damage could be observed in acute stage upon radiation at this dosage we chose. Therefore, we only analyzed one sample and cannot quantify it. The repeatability of the remaining experimental operations in this article is 3.
- L227 - Targeted Brain and Plasma Lipidomics: Among all lipids measured only two appeared to be significantly modulated in brain or in plasma; how do you explain the differences with the non-targeted lipidomics? It would be probably more relevant to build the discussion on these latter results. Error bars are very large; what is the number of analyses for this analysis?
Response: Non-targeted and targeted lipidomics are different methods to screen differential lipids. Non-targeted lipidomics is qualitative analysis, and targeted lipidomics is quantitative analysis of the lipids. Qualitative lipidomics is used first to build a profile of the modified lipids. A second, quantitative analysis targets the lipids identified during qualitative analysis.
It is worth noting that we only used internal standards to quantify 13 DAGs and 6 TAGs with significant differences that were screened in untargeted lipidomics and showed in Figure 3. Due to the small sample size, we only performed 3 biological replicates per group, which resulted in an excessively large error bar.
Although the error bars between groups are larger than expected, the robustness of the findings indicate to us that we are observing a real effect. We have decided to revise the manuscript with a cogent explanation to avoid further misunderstandings. At the same time, we appropriately adjusted the paragraph order of Discussion to make it more logical to read. In the revised manuscript, L314-L326 is a discussion of the results of untargeted lipidomics, and L327-L346 is a discussion of the results of targeted lipidomics. Please see for details in the manuscript highlighted in Discussion.
Our aim is to reveal the changes in lipid levels in the acute phase of brain injury caused by whole brain carbon heavy ion radiation. Lipidomic-detection helps us understand the changes in lipids directly related to radiation brain damage, and how lipids affect brain structure and function. Lipidomic-detection in plasma can be used as a potential biomarker for radiation brain damage.
We first conducted untargeted lipidomics analysis to build up the differential lipids. Among the 3,031 lipids detected by UPLC-MS, phosphatidylserine (PS), phosphatidylinositol (PI) and cholesteryl ester (CE) were detected but not significantly dysregulated upon heavy ion radiation. GPs and GLs were the most preferentially modified lipids. GPs are key components of membrane structure, stability, fluidity, and permeability, and are essential for membrane proteins, receptors, and ion-channels function [42-44]. DAGs are important precursors for the formation of various phospholipids, and participate in the metabolic cycle of lipids and hormones as an intermediate product of lipid metabolism [45, 46]. In addition, DAGs act as a secondary messenger, phosphorylating a range of substrate proteins by activating protein kinase C (PKC) [47]. On the other hand, TAGs provide a major source of energy and constitute a critical component of the lipoprotein [48]. Our untargeted lipidomics studies revealed that the contents of DAGs and TAGs in the irradiated group were significantly down-regulated compared to controls, indicating that differential lipids might be dysregulated upon irradiation (Figure 3).
DAGs can be phosphorylated to phosphatidic acid (PA) by diglyceride kinase (DGK) to further regulate cell function [49]. There are five subtypes of DGK, among which DGKβ is expressed in neuronal cells and plays an important role in brain function [50, 51]. Studies have shown that DGKβ knockout mice exhibit cognitive impairment and anxiety [52, 53]. TAGs can promote the transport of peripherally derived peptides ghrelin and insulin that have a positive effect on cognition in the blood-brain barrier [54]. Considering the functional relevance of DAGs and TAGs and based on the results of untargeted lipidomics, we conducted targeted quantitative analysis of 19 DAGs and TAGs that showed differences (Figure 4A and Figure 4B) in both brain tissue and plasma. Overall, 11 out of 19 DAG and TAG differential lipids were detected and quantified in both brain and plasma probably due to the small number of samples. Combined with the changes in the relative abundance of each DAGs and TAGs lipid in the control group and the radiation group in Figure 4, we found the contents of DAG 47:1, DAG 53:1 and DAG 50:6e in the rat brain tissue negatively correlated with those in the plasma, possibly due to the release of DAG 47: 1, DAG 53: 1 and DAG 50: 6e from the damaged brain tissue to the blood plasma. Among them, DAG 47:1 in the brain tissue post irradiation showed significant down-regulation in both untargeted and targeted lipidomics, suggesting that DAG 47:1 may be a lipid directly related to radiation brain damage. The results of targeted lipidomics in plasma showed that the relative content of DAG 47:1 in the irradiated group had an up-regulation trend, indicating that DAG 47:1 could be used as a potential biomarker of radiation brain damage. However, due to the small sample size in this experiment, the error bar is too large, so no significant trend of change is seen.

Reviewer 2 Report
Manuscript by Li et al. entitled “Effect of Heavy Ion 12C6+ Radiation on Lipid Constitution in the Rat Brain” is a revised version of a previously submitted manuscript “Effect of Heavy Ion Metal Radiation on Lipid Constitution in the Rat Brain” and describes changes in the lipid content of the brain and blood plasma of rats exposed to an acute dose of 15 Gy of 12C6+ ions.
As a result of the previous review cycle, the authors carried out the suggested measurements of the lipids in the blood plasma, since it is the blood sample that is typically used for biomonitoring or triage purposes in cases of accidental exposures of humans to radiation at intermediate to high doses. Also, histological evaluation of the cortex and hippocampal brain regions was done additionally. The authors are commended for carrying out additional experiments. However, these results do not resolve the major deficiencies identified before: 1) the lack of a lower dose 2) the lack of relevance to real-life exposure scenarios and to prediction of cognitive impairment.
Indeed, no experiments with a lower dose, e.g. a dose of 2 Gy, have been done. The dose of 2 Gy, as cited by the authors, is the highest acute dose that can be experienced by astronauts during the solar particle events and while being in open space – an extremely rare event that has never occurred in history of human space exploration. Also, in a such scenario, it is protons, not heavy ions, that contribute most to the absorbed dose. Therefore, the use of a lower dose is not an option for this study to qualify for publication within the relevance context chosen by the authors. In regard to this, the authors write “In this study, a 12C6+ heavy ion beam with a total of 15 Gy (260 MeV, LET = 13.9 KeV/μm) low LET plateau irradiation was used to irradiate the whole rat brain to simulate the accumulated strength of long term radiation.” It is commonly appreciated that dose-rate is a significant factor in defining the health related outcome of irradiation, therefore, claiming that acute dose of 15 Gy is a simulation of a total dose accumulated during a long term exposure is inappropriate.
Furthermore, the presented data (images only) from histological assessment of damage to the brain are not acceptable since they lack quantification and the presented images lack visual differences between the Control and Irradiation as judged by a neurobiologist researcher in the laboratory of this Reviewer. Indeed the authors write “However,in the irradiated group (Figure 1B), there were more neuron shrinkage in the cerebral cortex. The cell staining was deeper. The nucleus and cytoplasm were not clearly defined. There was no significant difference in hippocampus between the irradiated group and the control group. Pathological analysis showed that 12C6+ irradiation did cause brain damage in the rat brain.” It is clear that such claims are unsupported by the presented images lacking quantification.
Although the authors did report on the recommended measurements of blood plasma levels of DAG 47:1, that was substantially reduced in the irradiated brains, the results showed the lack of change (although the authors unjustifiably present it as a trend in the opposite direction). Instead the upregulation of DAG 53:1 was found in the plasma, in contrast to the brain where it did not change. No proper discussion of such results is provided in the manuscript. Although the authors rightly pointed that the directionality of changes in the brain and in the plasma were opposite, their claim that ”This obvious differential amount of DAG 47: 1, DAG 53: 1 and DAG 50: 6e in the rat brain tissues and blood plasma after irradiation indicate that they can be potentially used as biomarkers of cognitive dysfunction caused by heavy ion irradiation” This claim lacks experimental evidence and is poorly justifiable. In this Reviewer’s opinion, it is only DAG 47:1 has a potential for use in predicting neurological impairment in astronauts using blood samples, only if the result is confirmed for doses of 2 Gy and below.
In conclusion, the manuscript cannot be recommended for a publication in its present form. Additional experiments using a lower dose, e.g. 2 Gy are required for the study to generate knowledge relevant to any human exposure situations.
Author Response
As a result of the previous review cycle, the authors carried out the suggested measurements of the lipids in the blood plasma, since it is the blood sample that is typically used for biomonitoring or triage purposes in cases of accidental exposures of humans to radiation at intermediate to high doses. Also, histological evaluation of the cortex and hippocampal brain regions was done additionally. The authors are commended for carrying out additional experiments. However, these results do not resolve the major deficiencies identified before: 1) the lack of a lower dose 2) the lack of relevance to real-life exposure scenarios and to prediction of cognitive impairment.
Indeed, no experiments with a lower dose, e.g. a dose of 2 Gy, have been done. The dose of 2 Gy, as cited by the authors, is the highest acute dose that can be experienced by astronauts during the solar particle events and while being in open space – an extremely rare event that has never occurred in history of human space exploration. Also, in a such scenario, it is protons, not heavy ions, that contribute most to the absorbed dose. Therefore, the use of a lower dose is not an option for this study to qualify for publication within the relevance context chosen by the authors. In regard to this, the authors write “In this study, a 12C6+ heavy ion beam with a total of 15 Gy (260 MeV, LET = 13.9 KeV/μm) low LET plateau irradiation was used to irradiate the whole rat brain to simulate the accumulated strength of long term radiation.” It is commonly appreciated that dose-rate is a significant factor in defining the health related outcome of irradiation, therefore, claiming that acute dose of 15 Gy is a simulation of a total dose accumulated during a long term exposure is inappropriate.
Response:We selected low LET plateau irradiation with a small deposition dose (260 MeV, LET = 13.9 KeV/μm). The plateau region of heavy ion beam and X-rays belong to low linear energy (LET) radiation. The radiation biological effect produced by the plateau region of heavy ion beam is much lower than the radiation biological effect produced in or near the Bragg peak area. Again, anything that could happen in high doses could also theoretically occur at low doses, but we may have to introduce high doses to see the effects more clearly, resulting in more-accurate biomarker inventions.
Our team’s previous study (Zhang et al., 2019) on the sensitivity of rat brain regions under gamma radiation showed that the weight of rats dropped significantly on the 7th day after a single dose of 30 Gy of whole brain gamma irradiation, and then began to gradually recover, and the mortality of rats after irradiation is zero. At the same time, it was observed that the immune function and hematopoietic function of the irradiated rats were acutely damaged, which was manifested as a significant decrease in the content of white blood cells and platelets in the peripheral blood. Based on our research experience, it was suggested that the biological effects of radiation caused by carbon heavy ion irradiation in the plateau at the same dose are about twice that of gamma irradiation. In addition, we previously constructed the rat model of brain-localized 15 Gy carbon ion radiation (plateau region) and the rats can survive for more than three months after irradiation (Lei et al., 2015). Based on the above research experience, we continued to choose the dose and time point 7 days (acute phase) after 15 Gy whole brain irradiation with carbon heavy ions in the plateau as the irradiation conditions for this experiment.
According to the reviewer’s suggestion and after discussions with researchers in the field of radiation biology, we revised the background of the manuscript to avoid ambiguity. Please see below for details in the manuscript highlighted in Introduction and Discussion.
Introduction: In biomedical research, heavy ion beam currents are used to investigate the biological effects of ground-based simulated cosmic rays. They are also commonly used in radiotherapy to treat cancer patients [1]. Unlike X-ray and gamma radiation, heavy ions have a reverse depth dose distribution and high relative biological effect (RBE), making them ideal tools for head, neck, and central nervous system tumors [2]. Furthermore, cranial radiotherapy remains the mainstay of palliative treatment in patients with multiple symptomatic brain metastases [3]. Radiotherapy does not only target the tumors, but also affects the healthy brain tissue surrounding the tumor. However, research on the effect of radiotherapy on healthy brain tissue is limited. Studies have shown that heavy ion radiation causes structural and functional damage in the CNS regardless of low or high LET irradiation [4, 5]. Clinically, CNS damage caused by whole brain irradiation is divided into acute (≤7d), early-(up to months) and late-delayed stages (several months to years) [6]. Brain histopathology at the late-delayed stage following medium or high irradiation with heavy ions showed irreversible cerebral spinal cord disorders including vascular injury and white matter necrosis. Behavioral consequences of spinal cord disorders include cognitive dysfunction and abnormal walking patterns [7, 8]. In contrast, the pathological and behavioral consequences of the acute stage following irradiation are more subtle [8]. Non-invasive, early onset, biomarkers have been developed in response to heavy-ion radiation damage mediated cognitive decline.
Discussion: While heavy ion radiation is widely used to treat cancer patients, it also induces more serious biological damage than X- or gamma-ray radiation due to the sharp high-energy Bragg peaks [28, 29]. Carbon ions have become a star beam for effective treatment of brain tumors and melanomas due to their high LET, high precision and low side effects [30]. Due to the advancement of system therapy and radiotherapy equipment technology in recent years, the survival period of radiotherapy patients is getting longer and longer. However, the late effects of radiation brain injury, including cognitive impairment, have enough time to manifest and seriously threaten the patient’s prognostic quality of life. Traditional radiotherapy is 1.5-2 Gy in daily fractions, and the total treatment lasts 5-6 weeks to reach a total dose of 50-70 Gy [31]. However, modern radiotherapy is rapidly changing the standard of cancer treatment. For example, extreme oligofractionation is high-dose, single-fraction treatment [32]. The National Institute of Radiology (NIRS) of Japan uses a single delivered dose of 25-30 Gy carbon ion beam to treat stage I/II non-small cell lung cancer [33]. Compared with fractional radiotherapy, a single high-dose radiotherapy has demonstrated a greater therapeutic advantage in limited clinical trials. However, damage to healthy tissues around the lesion and along the path of radiation is still a factor that needs to be considered during radiotherapy. Herein, we aim to find potential biomarkers of radiation brain injury can be discovered as early as possible, so that early intervention and treatment of cognitive dysfunction that may occur in patients in the later period can be carried out.
Furthermore, the presented data (images only) from histological assessment of damage to the brain are not acceptable since they lack quantification and the presented images lack visual differences between the Control and Irradiation as judged by a neurobiologist researcher in the laboratory of this Reviewer. Indeed the authors write “However, in the irradiated group (Figure 1B), there were more neuron shrinkage in the cerebral cortex. The cell staining was deeper. The nucleus and cytoplasm were not clearly defined. There was no significant difference in hippocampus between the irradiated group and the control group. Pathological analysis showed that 12C6+ irradiation did cause brain damage in the rat brain.” It is clear that such claims are unsupported by the presented images lacking quantification.
Response:We have revised the figures and the explanation of the results in text after discussion with two other neurobiologists who have experience in brain research. We added general view and partial view of the hippocampus CA area and DG area in Figure 1, increased the magnification of the pictures, and displayed the scale bar value under each picture. Please see below for details in the manuscript highlighted from L112 to L132 in section 2.2. Based on our previous study, we did not expect obvious damage could be observed upon radiation at this dosage we chose. Therefore, we only analyzed one sample and cannot quantify it.
“Radiation-induced cognitive dysfunction frequently occurs in the hippocampus and prefrontal cortex [27, 28]. In order to determine the effects of 12C6+ irradiation on the rat brain, we randomly selected a rat's left hemisphere from each group to make paraffin sections and H&E staining analysis. Figure 1A and Figure 1B represent the general views of the left hemispheres of rats in the control and irradiation groups, respectively (bar =1000 μm). Figure A1 and Figure B1 represent the general view of the prefrontal cortex of rats in the control group and the irradiated group after magnification of 3 times (bar = 500 μm). The cell morphology of the prefrontal cortex of rats in the control group and the irradiated group after 40 times magnification (bar = 20 μm) was checked as shown in Figure A4 and Figure B4. Compared with the control group, the neuron nuclei in the prefrontal cortex of the irradiated rats shows vacuolation (black arrow in Figure B4), but no inflammatory infiltration can be observed. Pathological analysis of the hippocampus, another sensitive area of brain radiation damage was also conducted. Figure A2 and Figure A3 represent the general view of the cornu ammonis (CA) area and dentate gyrus (DG) area of the hippocampus of the control group, respectively (bar = 100 μm). Figure A5 and Figure A6 are the partial views of the CA area and DG area after 40 times magnification (bar = 20 μm). Compared with Figure B2 and Figure B3 (bar = 100 μm), which also represent the CA area and DG area of the irradiated group, the neurons in the hippocampus are abundant and closely arranged in both control and irradiated group. The neuron morphology is normal and the nucleolus is obvious. Figure B5 and Figure B6 can be seen in enlarged views of the CA and DG areas of the hippocampus in the irradiated group (bar = 20 μm). The results of the above pathological analysis show that the acute period after 15 Gy 12C6+ irradiation did cause morphology changes on cells, although not obvious in the rat brain after 7 days.”
Although the authors did report on the recommended measurements of blood plasma levels of DAG 47:1, that was substantially reduced in the irradiated brains, the results showed the lack of change (although the authors unjustifiably present it as a trend in the opposite direction). Instead the upregulation of DAG 53:1 was found in the plasma, in contrast to the brain where it did not change. No proper discussion of such results is provided in the manuscript. Although the authors rightly pointed that the directionality of changes in the brain and in the plasma were opposite, their claim that ” This obvious differential amount of DAG 47: 1, DAG 53: 1 and DAG 50: 6e in the rat brain tissues and blood plasma after irradiation indicate that they can be potentially used as biomarkers of cognitive dysfunction caused by heavy ion irradiation” This claim lacks experimental evidence and is poorly justifiable. In this Reviewer’s opinion, it is only DAG 47:1 has a potential for use in predicting neurological impairment in astronauts using blood samples, only if the result is confirmed for doses of 2 Gy and below.
Response:Our previous study on the sensitivity difference of brain regions under whole brain radiation exposure showed that a single 5 Gy (260 MeV, LET = 13.9 KeV/μm) plateau region of 12C6+ radiation did not cause any observable damage. Anything that could happen in high doses could also theoretically occur at low doses, but we may have to introduce high doses to see the effects more clearly, resulting in more-accurate biomarker inventions.
Due to the small sample size, we only performed 3 biological replicates per group, which resulted in an excessively large error bar. Although the error bars between groups are larger than expected, the robustness of the findings indicate to us that we are observing a real effect. We have decided to revise the manuscript with a cogent explanation to avoid further misunderstandings. At the same time, we appropriately adjusted the paragraph order of Discussion to make it more logical to read. In the revised manuscript, L327-L346 is a discussion of the results of targeted lipidomics. Please see for details in the manuscript highlighted in Discussion.
“DAGs can be phosphorylated to phosphatidic acid (PA) by diglyceride kinase (DGK) to further regulate cell function [49]. There are five subtypes of DGK, among which DGKβ is expressed in neuronal cells and plays an important role in brain function [50, 51]. Studies have shown that DGKβ knockout mice exhibit cognitive impairment and anxiety [52, 53]. TAGs can promote the transport of peripherally derived peptides ghrelin and insulin that have a positive effect on cognition in the blood-brain barrier [54]. Considering the functional relevance of DAGs and TAGs and based on the results of untargeted lipidomics, we conducted targeted quantitative analysis of 19 DAGs and TAGs that showed differences (Figure 4A and Figure 4B) in both brain tissue and plasma. Overall, 11 out of 19 DAG and TAG differential lipids were detected and quantified in both brain and plasma probably due to the small number of samples. Combined with the changes in the relative abundance of each DAGs and TAGs lipid in the control group and the radiation group in Figure 4, we found the contents of DAG 47:1, DAG 53:1 and DAG 50:6e in the rat brain tissue negatively correlated with those in the plasma, possibly due to the release of DAG 47: 1, DAG 53: 1 and DAG 50: 6e from the damaged brain tissue to the blood plasma. Among them, DAG 47:1 in the brain tissue post irradiation showed significant down-regulation in both untargeted and targeted lipidomics, suggesting that DAG 47:1 may be a lipid directly related to radiation brain damage. The results of targeted lipidomics in plasma showed that the relative content of DAG 47:1 in the irradiated group had an up-regulation trend, indicating that DAG 47:1 could be used as a potential biomarker of radiation brain damage. However, due to the small sample size in this experiment, the error bar is too large, so no significant trend of change is seen.“

Round 2
Reviewer 1 Report
The manuscript has gone significant revision with a positive impact on the justifications and aims of the study. The manuscript has been improved in many parts; but some repetitions are present along the text and needs to be avoided and some parts need to move. A few comments are enclosed in this sense.
L64: one reference is missing (after “necrosis”)
L130 (part 2.1), just add the statistical test used for these results or refer to “Statistical analysis”
L294: Figure 1: add “n=1”
L 338: Remove this part that is already written in the material and methods section “UPLC-ESI-TOF-MS data was processed using the MS-DIAL 3.96 data analysis software [26]. The analysis process was sequentially performed, from alignment, peak picking, and lipid identification. The Lipid Maps Database (www.lipidmaps.org) was used to identify metabolites.”
L 344: Put the following part in the material and methods section “PCA is an unsupervised data analysis tool used to reduce and visualize multidimensional data (X) without referring to class labels (Y) The data was summarized into fewer variables called principal components, and weighted average of the original variables.”
L360: “asstatistically”
L428: “accounted for”
L503: check why the term “relative” level?
L533-l548 should probably be inserted in introduction instead of discussion of results.
L985 O2 generates
L1004 “Studies have shown that radiations damages the CNS microvasculature and cause microglia”
In conclusion: “for pharmacotherapeutic intervention to provide radiation protection”. The link with radiation protection is not so obvious.
Author Response
Comments and Suggestions for Authors
The manuscript has gone significant revision with a positive impact on the justifications and aims of the study. The manuscript has been improved in many parts; but some repetitions are present along the text and needs to be avoided and some parts need to move. A few comments are enclosed in this sense.
L64: one reference is missing (after “necrosis”)
Response: According to the reviewer’s suggestions, we added references to the revised manuscript with blue highlight.
L130 (part 2.1), just add the statistical test used for these results or refer to “Statistical analysis”
Response: According to the reviewer’s suggestions, we have marked significant differences in Table 1 and Table S1 to indicate whether there is significant difference between the control group and the irradiation group.
L294: Figure 1: add “n=1”
Response: We have added “n=1” in the legend. According to the reviewer 2’s suggestion, we remove Figure 1 to the supplementary information, and concise the description of the results in section 2.2. Please see below for details in the manuscript highlighted in section 2.2.
“Radiation-induced cognitive dysfunction frequently occurs in the hippocampus and prefrontal cortex [25, 26]. In order to determine the effects of 12C6+ irradiation on the rat brain, we randomly selected a rat's left hemisphere from each group to make paraffin sections and H&E staining analysis (see supplementary information Figure S1). Compared with the control group, the neuron nuclei in the prefrontal cortex of the irradiated rats showed vacuolation (black arrow in Figure S1B4), but no inflammatory infiltration could be observed. The results of pathological analysis of the hippocampus, another sensitive area of brain radiation damage, showed that the neurons were abundant and closely arranged, with normal neuronal morphology and obvious nucleolus in both control and irradiated group. The above results indicated that the acute period after 15 Gy 12C6+ irradiation did cause morphology changes on cells, although not obvious in the rat brain after 7 days. ”
L 338: Remove this part that is already written in the material and methods section “UPLC-ESI-TOF-MS data was processed using the MS-DIAL 3.96 data analysis software [26]. The analysis process was sequentially performed, from alignment, peak picking, and lipid identification. The Lipid Maps Database (www.lipidmaps.org) was used to identify metabolites.”
Response: We have revised the manuscript according to the reviewer’s suggestion.
L 344: Put the following part in the material and methods section “PCA is an unsupervised data analysis tool used to reduce and visualize multidimensional data (X) without referring to class labels (Y) The data was summarized into fewer variables called principal components, and weighted average of the original variables.”
Response: We have revised the manuscript according to the reviewer’s suggestion.
L360: “asstatistically”
Response: It was a typing error and we revised it.
L428: “accounted for”
Response: We have revised it according to the reviewer’s suggestion.
L503: check why the term “relative” level?
Response: Standard deviation (SD) is a measure of the degree to which a set of values are scattered from the average. The "relative levels” mentioned here are the relative comparison to the average of each type of lipid.
L533-L548 should probably be inserted in introduction instead of discussion of results.
Response: We could not locate L533-L548 in the discussion section. L533-L548 in the version downloaded from the system is reference. Therefore, we did not change it.
L985 O2 generates
Response: We have revised it according to the reviewer’s suggestion.
L1004 “Studies have shown that radiations damages the CNS microvasculature and cause microglia”
Response: We have revised it according to the reviewer’s suggestion.
In conclusion: “for pharmacotherapeutic intervention to provide radiation protection”. The link with radiation protection is not so obvious.
Response: We have revised the conclusion. Please see below for details in the manuscript highlighted in conclusion.
“The availability of lipid profiles will expand our understanding on the consequences of heavy ion irradiation and lay the foundation for subsequent exploration of the molecular mechanisms of radiation brain damage.”

Reviewer 2 Report
The authors of “Effect of Heavy Ion 12C6+ Radiation on Lipid Constitution in the Rat Brain” are commended for following the advice and changing the context in which the results should be presented and discussed – from one related to space radiation protection to the context of brain radiotherapy. Justification for the use of a high dose of 15Gy is now valid (and reinforced by the added new text and references) and results present interest to the medical neurobiologists community.
Once last comment that this Reviewer must make concerns the histological images shown in Fig.1. First, the Figure legend and materials and methods should explicitly say that only one animal per treatment was used, otherwise this portion can be misleading. Second, it is highly recommended that this data - coming from only one biological replicate – be placed in supplementary materials, due to its limited value for conclusiveness.
Overall, the revised manuscript is sufficiently improved and is recommended for publication.
Author Response
Comments and Suggestions for Authors
The authors of “Effect of Heavy Ion 12C6+ Radiation on Lipid Constitution in the Rat Brain” are commended for following the advice and changing the context in which the results should be presented and discussed – from one related to space radiation protection to the context of brain radiotherapy. Justification for the use of a high dose of 15Gy is now valid (and reinforced by the added new text and references) and results present interest to the medical neurobiologists community.
Once last comment that this Reviewer must make concerns the histological images shown in Fig.1. First, the Figure legend and materials and methods should explicitly say that only one animal per treatment was used, otherwise this portion can be misleading. Second, it is highly recommended that this data - coming from only one biological replicate – be placed in supplementary materials, due to its limited value for conclusiveness.
Overall, the revised manuscript is sufficiently improved and is recommended for publication.
Response: We have revised the manuscript according to the reviewer’s suggestion. We move Figure 1 in the supplementary information (Figure S1), and modified section 2.2. We clearly indicate that only one animal per treatment was used in Materials and Methods 4.2 and in the Figure S1 legend. Please see below for details in the manuscript highlighted in section 2.2.
“Radiation-induced cognitive dysfunction frequently occurs in the hippocampus and prefrontal cortex [25, 26]. In order to determine the effects of 12C6+ irradiation on the rat brain, we randomly selected a rat's left hemisphere from each group to make paraffin sections and H&E staining analysis (see supplementary information Figure S1). Compared with the control group, the neuron nuclei in the prefrontal cortex of the irradiated rats showed vacuolation (black arrow in Figure S1B4), but no inflammatory infiltration could be observed. The results of pathological analysis of the hippocampus, another sensitive area of brain radiation damage, showed that the neurons were abundant and closely arranged, with normal neuronal morphology and obvious nucleolus in both control and irradiated group. The above results indicated that the acute period after 15 Gy 12C6+ irradiation did cause morphology changes on cells, although not obvious in the rat brain after 7 days. ”

This manuscript is a resubmission of an earlier submission. The following is a list of the peer review reports and author responses from that submission.
Round 1
Reviewer 1 Report
The major focus of the manuscript presented by Bo Li et al. was set on the effect of Heavy Ion Metal Radiation on lipid constitution in the rat brain. These effects deal with lipid distribution pattern and oxidative stress marker assessment in rat brain with or without heavy ion irradiation.
Overall the topic of the manuscript is highly interesting, and may display a preliminary study for an increased mechanistic understanding of lipid damage following heavy ion radiation exposure. Suggesting pathway is interesting. However, there are a set of significant issues. Major and minor points of criticism are given successively.
Minor points:
- Title: “Effect of Heavy Ion Metal Radiation on lipid constitution in the rat brain”: Remove metal, insert the 12C6+ in the title.
- L36-37: As you cite this example, could you please mention the dose rate due to heavy ion in Space?
- L42-43: “early central nervous system dysfunction caused by radiation cannot be judged by behavioral performance” -> explain why?
- L69-70: Data of body weight needs to be given at least in the mat/meth (or in supplementary)
- Table1: Insert in the caption the statistical tests associated to these figures. In the column head, “g” unit needs to be into brackets.
- Figure 3A, B, C: increase the font of x-axis description words.
Major points:
- Animal experimental authorization: Was the animal experimental procedure assessed by an ethical committee? Please give the reference. Could you please give also the agreement number of the animal facility?
- In the material and methods, the number of rats in each group is 8 but in the results, only 3 results are shown. Could you please explain these differences? What about the 5 others per condition?
- L63: Choice of the dose: 15Gy seems huge; could you please explain your choice? In addition, explain the choice of irradiation mode: acute high dose instead of for instance moderate repeated doses or chronic.
- The choice of 7 days post-irradiation analysis should be more argued in a biological point of view (particular step of developmental process?).
- Lipidomic results: a/ one table with all results (name of lipids, fold change, pvalue) is missing; it can be given in supplementary data but needs to be added.
b / Please comment the number of lipids found; are they really 3497 lipids or 3497 peaks? Is 3497 important or not? Compare with previous studies.
c/ Lipidomics enables to determine the changes of relative lipid distribution between two conditions; what about the absolute amount of lipids in each condition? Is it stable? Did you assess total neutral and polar lipids?
d/ Usually, non-quantitative lipidomics is used to perform a first screening of main modified lipids but a second step with the main hit is necessary to perform to get a quantitative confirmation. Such validation is necessary to confirm the hypothesis of mechanisms. Have you done it? Please mention what is planned about this step.
- Figure 1: results of PCA should be useful to add for comparison, above all in order to find the main influencing components (differential lipids). In the caption, why do you talk about metabolomics positive ionization mode? It is confusing.
- L105-106: a/ explain precisely how did you construct the different groups and cite the different lipids in each group (new table in the document or in supplementary). b/ If I understood well, this work has only been done on the significantly modified lipids; what is the distribution of the non-modified lipids through the different classes, to be able to assess the proportion of modified/non-modified lipids in each class?
- L180: the discussion seems indicating that lipid damage studies are useful to detect changes before behavior differences but how do you make sure that both are linked? The link between lipid damages and behavior trouble should be illustrated with a reference. Are lipid changes irreversible? A biological proof of damage would have been useful (histological for instance).
- Results on MDA and SOD should be better explained and commented: Heavy ions are they really likely to produce ROS? Illustrate the process in comparison with, for instance, gamma production of ROS (or give a relevant reference)? Is MDA really a good marker of lipid peroxidation and lipid damage?
- For the discussion on heavy ion irradiation, please indicate how much penetrate heavy ions in the biological tissue and more specifically in brain?
Reviewer 2 Report
Manuscript by Han et al. entitled “Effect of Heavy Ion Metal Radiation on Lipid Constitution in the Rat Brain” describes results of a small study looking into changes in the lipid content of the brain of rats exposed to heavy ion radiation. The authors exposed rats to a single whole head dose of 15 Gy of 12C ions and extracted brains 7 days post-irradiation for lipid measurements by UPLC-MS. They identified DAG47:1 as the most affected molecule and proposed its use as a biomarker of brain toxicity upon irradiation before behavioral changes can occur. The authors also proposed a mechanism of heavy ion radiation-induced alterations to lipid metabolism based on the analysis of the lipid profile changes and MDA level and SOD activity changes in the brain.
Whereas this Reviewers has no major concerns about the technical performance of the study, the following significant flaws do not allow this work to be recommended for publishing.
Relevance: This study has no relevance to any of the human exposure scenarios mentioned by the authors in the introduction or abstract. They spoke of human space travel, however, dose received by astronauts are 2 orders of magnitude lower (Cucinotta et al. Space Radiation Cancer Risk Projections and Uncertainties - Available from: https://ntrs.nasa.gov/search.jsp?R=20130001648.) Further, the authors make strong link to radiotherapy of cancer, however, off-target dose to healthy tissues received during heavy-ion radiotherapy are at least one order of magnitude lower (Jarlskog et al. 2018. Phys Med Biol). Results do not support the claims: the authors' claims are based on results obtained from only a single experiment using only one dose (which is much greater than any relevant human exposure scenario) and only one timepoint. Scientific significance: is low not only because of the two points above, but also because of the fact that such a high dose of radiation (equivalent dose to brain will be 15 Gy x RBE for Cardon ions) triggers severe damage and will cause behavioral changes and possibly death of animals. Further, the authors claimed that DAG47:1 may be used as a biomarker; what is a use of a biomarker that requires a brain sample for measurements and therefore the death of an organism?
The study would improve if the authors: 1) carried out lower-dose irradiations, e.g. down to 1 or 0.5 Gy, relevant to human exposure scenarios; 2) measured DAG47:1 (and/or other affected molecules) in blood plasma (which was easy to do in this experiment); 3) monitored behavioral/cognitive changes in a sub-group of exposed rats to correlate identified markers.
There are other concerns relating to data presentation (some plots have unreadable labels, some plots refer to mice, rather than rats, etc.), using references inappropriately (e.g. references 4 and 5 have nothing to do with space radiation), lack of technical information on radiation quality (LET, energy), poor coverage of the relevant literature on lipids metabolism in the context of radiation.
In summary, given the described major weaknesses of the study making it irrelevant and of low scientific value, it is not recommended for publication. Substantial additional experiments, also suggested above, are required to improve the study.